# Positive Effect of a New Combination of Antioxidants and Natural Hormone Stimulants for the Treatment of Oligoasthenoteratozoospermia

**DOI:** 10.3390/jcm11071991

**Published:** 2022-04-02

**Authors:** Vincenzo De Leo, Claudia Tosti, Giuseppe Morgante, Rosetta Ponchia, Alice Luddi, Laura Governini, Paola Piomboni

**Affiliations:** 1Department of Molecular and Developmental Medicine, Siena University, 53100 Siena, Italy; vincenzo.deleo@unisi.it (V.D.L.); giuseppe.morgante@unisi.it (G.M.); ponchia2@student.unisi.it (R.P.); paola.piomboni@unisi.it (P.P.); 2Assisted Reproduction Unit, Siena University Hospital, 53100 Siena, Italy; claudia.tosti@ao-siena.toscana.it

**Keywords:** male infertility, oligoasthenoteratozoospermia, clinical-therapeutic strategies, antioxidant treatment

## Abstract

Oligoasthenoteratozoospermia (OAT) accounts for about 90% of male infertility; in many cases this disorder may be associated with oxidative stress, a condition that decreases the success of fertilization. Therefore, the empirical treatment of male infertility is often based on the use of antioxidants. The aim of the present study was to assess the effectiveness of three months’ administration of a new nutraceutical preparation on hormone profile, sperm parameters and fertilization capability in men undergoing in vitro fertilization (IVF). A total of 36 OAT patients were daily treated for 3 months with a dose of a formulation containing: Inositol, L-Carnitine, Vitamins C, D, E, Coenzyme Q10 and Selenium. Selected parameters were analysed before (T0) and after (T1) treatment, and IVF outcomes were evaluated. We observed an improvement of sperm concentration, motility, morphology and vitality; blood level of testosterone also showed an increase. A significant increase of fertilization rate was detected in 14 couples, whose male partner were treated with the nutraceutical preparation. The present results indicate that a formulation containing antioxidant and energy supply substances was effective in the treatment of sperm alterations and led to significant recovery of fertilizing capacity.

## 1. Introduction

The World Health Organization (WHO) estimates that in advanced industrial countries, couples with fertility problems constitute about 15–20% of the population of reproductive age, with the male factor contributing to almost half of the cases [1,2]. The problem seems to be increasing for reasons such as postponing parenthood, negative environmental factors, unhealthy life-styles and various social conditions.

Excluding anatomical defects, a low sperm count combined with poor sperm motility and morphology (oligoasthenoteratozoospermia, OAT) is considered one of the most common causes of male infertility. Of these, about 30% are unexplained; the others are linked to causes that range from hormonal alterations, genetic anomalies, iatrogenic factors and unhealthy life-style (diet, smoking, alcohol) [3,4,5] and oxidative stress by reactive oxygen species (ROS) [6]. In semen there is a homeostasis between free radicals, produced by leukocytes and spermatozoa [7], and protective enzymatic and non-enzymatic antioxidants. Some pathophysiology conditions, environmental factors or unhealthy lifestyles may alter this equilibrium and lead to an accumulation of ROS in seminal fluid, causing a harmful oxidative damage [8,9,10].

Regarding the hormonal component, infertile men are characterized by low plasma concentrations of testosterone and LH, almost invariably combined with a lack of energizing factors and antioxidants, all of which contribute to the onset of OAT.

With increasing recognition of the role of oxidative stress energy release in the pathophysiology of male infertility, the use of antioxidants/energizing compounds is one of the therapeutic options adopted for the treatment of idiopathic infertility. An ideal supplement should provide substances that affect plasma concentrations of testosterone, as well as energizing ingredients and antioxidants to improve sperm vitality and protect them against oxidative injury [11,12]. 

Many substances found in nature have such effects [5,13]. In particular, inositol seems to have specific effects on certain hormonal markers, favouring an increase in plasma concentrations of LH, followed by those of testosterone, through stimulation of Leydig cells in the testicles interstitium [14]. In vitro supplementation of myo-inositol is able to significantly improve sperm motility in a dose-dependent manner [15], demonstrating a protective role during sperm cryopreservation [16].

Another interesting substance is L-carnitine, which promotes energy and makes sperm more vital by improving their post-gonadic maturation. Arginine is also indicated as an amino acid useful for spermatogenesis and formation of nitric oxide, an energy source for sperm cells [17]. A major role among the antioxidants is played by the vitamin C and E, normally present in seminal fluid, which ensures the stability of cell structures and contributes to sperm motility [18]. Vitamin E (α-tocopherol) is an important lipid-soluble antioxidant molecule in the cell membrane. It is thought to interrupt lipid peroxidation and enhance the activity of various antioxidants that scavenge free radicals generated during the univalent reduction of molecular oxygen and during normal activity of oxidative enzymes [19]. The results of in vitro experiments suggest that vitamin E may protect spermatozoa from oxidative damage and loss of motility as well as enhance the sperm performance in the hamster egg penetration assay [20]. The ability of Vitamin C in suppressing the endogenous oxidative damage is also well documented [21]. Vitamin C concentration in the seminal plasma, is 10-fold higher than that in the serum [22]. And its levels in seminal plasma negatively correlated with the sperm DNA fragmentation index. Indeed, vitamin C supplementation has been reported improve sperm parameters in infertile men [23,24]. Vitamin D3 supplementation is increasingly accepted, since deficiencies have been correlated with the onset of certain male and female reproductive disorders. The enzyme responsible for vitamin D metabolism in the human sperm flagellum is correlated with the quality, vitality and function of mature sperm [25]. The antioxidant effects of coenzyme Q10 are well known, and although seminal fluid seems to protect sperm against oxidative stress, treatment with this coenzyme improves sperm motility [26,27]. Zinc is a fundamental trace element for reproductive processes, since it is involved in cell reproduction and protection against oxidative stress.

The present study aimed to investigate the impact of an oral antioxidant supplementation, composed of natural substances, on seminal and hormonal parameters of infertile men with OAT.

## 2. Materials and Methods

### 2.1. Study Design and Patients Recruitment

This prospective study was performed on a total of 36 Caucasian males undergoing semen evaluation at the Unit of Medically Assisted Reproduction, Siena University Hospital, after 12–18 months of unprotected sexual intercourse without conception. 

A comprehensive clinical history of patients was obtained; we excluded patients with possible causes of male infertility such as varicocele, cryptorchidism, endocrine disorders or systemic diseases and patients with intake of spermiotoxic drugs, smoking, alcohol or drugs abuse. All patients underwent microbiological analysis of seminal fluid and urine for common bacteria such as *Mycoplasma*, *Trichomonas vaginalis* and *Chlamydia trachomatis*. The median age of the patients was 34 years (range: 25–47 years); the BMI ranged between 18 and 25. All participants signed a written informed consent, and the study protocol was approved by the Ethic Committee of the Siena University Hospital (approval ID: CEASVE 191113).

In the selected patients, the diagnosis of oligoasthenoteratozoospermia (OAT) was confirmed by performing two spermiogram at one month (T0) from the first investigation (T-1). Recruited patients were asked to take orally, once a day, for three consecutive months, a preparation (Gomotil^®^, Gofarma, Italy) containing a cocktail of nutraceutical substances: Inositol, L-Carnitine, Acetyl L-Carnitine Hydrochloride, Vitamin E, Vitamin C, Coenzyme Q10, Selenium, and Vitamin D3 (Table 1).

At T0, the hormone profile of enrolled patients has been performed, including Testosterone, Follicle Stimulating Hormone (FSH), Luteinizing Hormone (LH), Sex hormone binding globulin (SHBG), Prolactin and Estradiol. The main blood-metabolic parameters were also measured: Glucose, Insulin, Creatinine, Total Cholesterol, Triglycerides, Oxaloacetic Transaminase, Pyruvic Transaminase and C-Reactive Protein. Sperm evaluation, hormone and metabolic profiles were tested after 3 months of treatment (T1; Table 2).

### 2.2. Semen Analysis

At T-1, T0 and T1 the ejaculate samples were collected, by masturbation, in sterile containers after a period of abstinence between 2 and 5 days. Semen samples were assessed according to WHO (2010) parameters [28]. 

Seminal analysis was carried out, within 30 min after fluidification. The volume, viscosity, pH, and appearance of the semen were evaluated together with sperm concentration, progressive and total motility, and morphology. Sperm concentration was evaluated using a Makler counting chamber (Irvine Scientific, Santa Ana, CA, USA), under an optical microscope (Nikon, Nikon Europe B.V., Amsterdam, The Netherlands) at 200× magnification. Sperm morphology was evaluated by using pre-coloured glasses (Testsimplets) and the eosin Y test was applied to evaluate sperm vitality. 

### 2.3. Assisted Reproduction Techniques

The main reproductive outcomes were retrospectively evaluated in 14 couples (Group A) undergoing IVF at the Unit of Medically Assisted Reproduction, Siena University Hospital, whose male partners received the nutraceutical supplementation for three months, before an IVF cycle. The control group (Group B) was composed by 14 couples where male, with a diagnosis of OAT, didn’t receive any supplementation before IVF.

Inclusion criteria were male infertility (OAT); exclusion criteria were female fertility factor and/or couple or idiopathic infertility. 

We included homolog cycles (no egg or sperm donors) using fresh oocytes and ejaculated sperm. Standard controlled ovarian stimulation protocols were used. Stimulation with gonadotrophins was monitored by measuring serum estradiol levels and follicle growth. Human chorionic gonadotropin was administered when patients reached the individual clinic’s trigger point for follicular growth. Cumulus–oocyte complexes were collected 36 h later, by ultrasound-guided transvaginal follicular aspiration. To perform ICSI, after 2 h of incubation the oocytes were denuded. Sperm injection was performed immediately after denudation according to conventional procedure. Fertilization was assessed 16–18 h after injection.

### 2.4. Statistical Analysis

A statistical analysis was performed by means of the GraphPad Prism 5.0 (GraphPad Software, San Diego, CA, USA) using nonparametric tests. The differences among groups of data, before (T0) and after (T1) the treatment with the nutraceutical preparation, were tested by the Kruskal–Wallis test. The data are reported as mean ± standard deviation (SD). The differences observed have been considered statistically significant at *p* < 0.05.

## 3. Results

### 3.1. Effect on Hormone and Metabolic Profile

The patients enrolled in the study did not have a history of endocrine, metabolic or anatomical alterations. Microbiological tests were all negative for Chlamydia trachomatis, Mycoplasma and Trichomonas vaginalis. 

The comparison of values before the beginning of the treatment (T0) and after three months (T1) shows an increase, although not statistically significant, of the blood levels of testosterone and in general an improvement of the hormonal profile (Table 3).

The effects of the nutraceutical preparation were evaluated at the T1 and compared with the T0 (Table 4). The analysis shows an improvement, even if not statistically significant, of the main metabolic parameters. This result, together with the absence of side effects, evidences the safety of the product making it suitable also for a large-scale use and for prolonged periods.

### 3.2. Effect on Sperm Parameters

The data analysis shows that the treatment with nutraceutical preparation significantly improve the main sperm parameters: a statistically significant rise of sperm concentration was evident, with an increase of 71.7% (* *p* < 0.05) (Figure 1A).

The total sperm number was increased of 28.6%, in a non-statistically significant manner (Figure 1B). Indeed, this parameter is closely related to contingent conditions such as ejaculate volume at the time of sample collection, which may be influenced by psychological stress or hydration of the patient. The effect of treatment on both progressive (Figure 1D) and total sperm motility (Figure 1E) showed an increase of 20.6% (* *p* < 0.05) and 19.6% (** *p* < 0.01) respectively. Sperm morphology evaluation demonstrated that the integrity and shape of the acrosome, the head morphology and the flagellum profile significantly changed after treatment with an increase of 61.5% (** *p* < 0.01; Figure 1F). Spermiogram data are summarised in Table 5. 

### 3.3. Reproductive Outcomes

At the end of 3 months of treatment with the nutraceutical preparation 14 patients (Group A) out of 36 underwent an IVF cycle at the UOSA PMA–University Hospital of Siena. The number of retrieved oocytes, MII injected oocytes, fertilization and pregnancy rate were registered and are reported in Table 6, in comparison with Group B.

## 4. Discussion

Male infertility is a significant social problem with a strong impact on well-being as well as an unbroken medical challenge. A large number of recent studies have focused on the ability of many substances, generally termed as *nutraceuticals*, to improve the hormonal status and sperm parameters by different mechanisms [29,30]. The supplementation with natural compound for the treatment of male infertility is greatly debated on literature. The evaluation of the effectiveness and safety of supplementary oral antioxidants in subfertile men put in evidence some important bias of the published studies, first of all the selection of patients and control groups. Results from observational studies might have been confounded by lifestyle factors such as age, weight, physical health and/or medication use. The administration of individual antioxidants or combinations of them, the dosage and formulation of the nutraceutical and the duration of treatment, can create discordant and non-significant results [11,31,32]. Apart from cases with a specific aetiology (genetic, hormonal, infectious etc.), which are readily diagnosed and treated medically and/or surgically, idiopathic alterations of the main sperm characteristics, as in the case of OAT, can benefit from the use of oral supplements based on amino acids (L-carnitine), antioxidants, such as vitamins A, C and E, folic acid and elements such as selenium [33,34]. This is not surprising, since oxidative stress resulting from an imbalance between ROS and antioxidants systems usually present in seminal fluid is fundamental in male fertility. Indeed, ROS abundance has been implicated in sperm abnormalities [35,36], while the exact impact on fertilization and pregnancy has long been the subject of considerable discussion. On the other end, reactive oxygen species mediate certain physiological processes such as sperm maturation, capacitation and acrosome reaction, two key events for the acquisition of fertilizing ability; therefore their fine balancing is fundamental to assure a proper redox microenvironment [37]. This is also supported by growing evidence demonstrating that the abuse of antioxidant treatments may induce sperm damage as a result of a reductive-stress-induced state. Therefore the phenomenon known as “antioxidant paradox” should be kept in mind and absolutely not underestimated in order to avoid that the uncontrolled supplementation may indirectly cause fertility health risks [38].

The results of this study show that the administration of a nutraceutical cocktail, containing amino acids able to provide energy, Vitamins C, D3 and E, Selenium and Coenzyme Q10 with a strong antioxidant action and other natural substances, can be considered an effective treatment for OAT men, showing a significant increase in all sperm parameters and thus suggesting a recovery of the fertilizing capability. 

The effects of the antioxidant therapy on seminal fluid have been studied in many clinical trials that have demonstrated individual and synergic action of compounds used in the nutraceutical formulation administered in this study.

Indeed, vitamin E and coenzyme Q10 have been reported to be effectives in protecting sperm against oxidative stress in cases of idiopathic infertility [34]. Several studies demonstrated that vitamin D3 plays key roles in the acquisition of hyperactivated motility, capacitation, and acrosome reaction. Despite these reports, there is no unanimous agreement on the effectiveness of vitamin D administration in recovering poor semen parameters. Indeed, some authors reported a beneficial effect of supplementation with vitamin D on sperm progressive motility and morphology in men with OAT, while others did not [39,40]. 

Carnitine is a key antioxidant involved in cell energy production, thus directly involved in recruiting ATP for sperm motility. To this regard, men with OAT have significantly lower levels of carnitine in their semen [34]. The combination of carnitine and acetyl-L-carnitine is effective in improving total motility in idiopathic asthenozoospermia [41]. 

Last but not the least, the most abundant component of the mix we used in this study is Inositol, whose effectiveness in improving sperm motility and morphology, along with a significant protective role against oxidative damage to DNA has been already demonstrated [15,16]. Data form literature show the beneficial effects of inositol on sperm motility and mitochondrial function, due to insulin-sensitizing properties, antioxidant activity and hormonal regulatory effects [42].

The effectiveness of this supplementation is definitively demonstrated by the positive effects on the fertilization rate, one of the most important parameter to evaluate the impact of male partner in the outcome of assisted reproduction cycles. Indeed, male partner received the aforesaid antioxidant supplementation for three months before the cycle obtained a higher fertilization rates obtained in ICSI cycles. The absence of side effects proves the safety of the product and provides specialists working in the field of assisted reproduction with alternative tools to classic hormonal therapies for the treatment of male infertility.

## 5. Conclusions

In conclusion, the present results demonstrate that a formulation containing amino acids as energy source, vitamin E and coenzyme Q10 with strong antioxidant effects, and natural substances that influence androgen production is an ideal therapy for OAT. 

Anyway, further studies in a larger cohort of patients are needed to confirm the effectiveness of this nutraceutical formulation in ameliorating sperm parameters. 

## Figures and Tables

**Figure 1 jcm-11-01991-f001:**
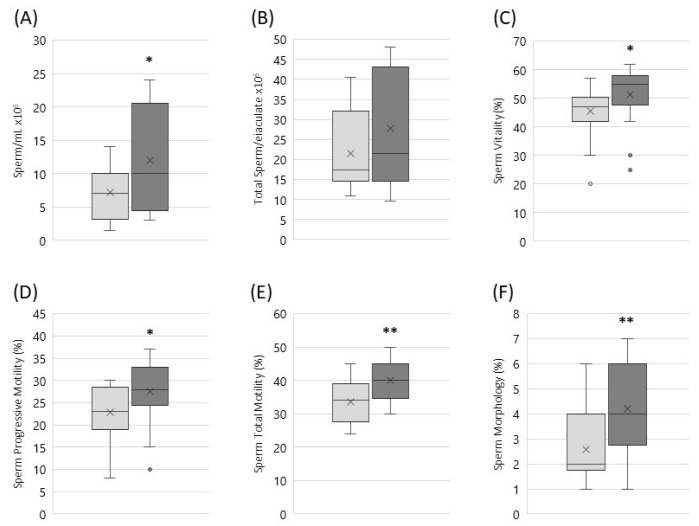
Sperm concentration (**A**); total sperm number (**B**); sperm vitality (**C**); progressive (**D**) and total (**E**) sperm motility; sperm morphology (**F**) before (T0; light grey) and after 3 months (T1; dark grey) treatment with nutraceutical preparation. Graphical diagrams are plotted as box–whisker plots, where boxes show the interquartile range with median and mean values, and whiskers represent min and max confidence intervals, outliers are represented as single dots (* *p* < 0.05; ** *p* < 0.01).

**Table 1 jcm-11-01991-t001:** Composition, Dosage and Nutrient Reference Values (NRVs) of the administered nutraceutical preparation.

Nutraceutical Composition	Per 1 Packet	*NRV*
**Inositol**	1000 mg	*-*
**L-Carnitine**	250 mg	*-*
**Acetyl L-Carnitine Hydrochloride**	250 mg	*-*
**Vitamin E**	60 mg	*500%*
**Vitamin C**	100 mg	*125%*
**Vitamin D3**	5 mcg	*100%*
**Coenzyme Q10**	20 mg	*-*
**Selenium**	50 mcg	*90.90%*

**Table 2 jcm-11-01991-t002:** Study Timeline.

Required Examination	T-1	T0 Basal	T1 Post Treatment
**Spermiogram**	X	X	X
**Hormonal panel**		X	X
**Hemato-Metabolic panel**		X	X

**Table 3 jcm-11-01991-t003:** Evaluation of the hormone levels before treatment with the nutraceutical preparation (T0) and after three months of administration (T1).

Test Parameters	T0 Basal	T1 Post Treatment	*Reference* *Values*
**Testosterone** (ng/mL)	4.5 ± 1.6	5.2 ± 1.8	*2.8–8.0*
**FSH** (mUI/mL)	5.2 ± 1.3	4.8 ± 0.9	*0.7–11.0*
**LH** (mUI/mL)	6.1 ± 1.7	5.5 ± 0.6	*0.8–8.0*
**SHBG** (nmol/mL)	48.0 ± 12.0	55 ± 16	*10.0–57.0*
**Prolactin** (ng/mL)	11.2 ± 3.0	10.6 ± 2.0	*2.0–13.0*
**Estradiol** (pg/mL)	32.0 ± 6.0	25.0 ± 4.0	*<32.0*

**Table 4 jcm-11-01991-t004:** Main blood values of patients before the beginning of the treatment with nutraceutical preparation (T0) and after three months of treatment (T1).

Test Parameters	T0 Basal	T1 Post Treatment	*Reference* *Values*
**Glucose** (mg/dL)	98 ± 7	91 ± 3	60–110
**Insulin** (microU/mL)	12.7 ± 4.2	9.3 ± 2.6	2.6–24.9
**Creatinine** (mg/dL)	0.88 ± 0.2	0.80 ± 0.3	*0.55–1.40*
**Total Cholesterol** (mg/dL)	218 ± 9	202 ± 5	*140–220*
**Triglycerides** (mg/dL)	160 ± 8	154 ± 7	*<200*
**Oxaloacetic Transaminase*****(AST)(GOT)*** (U/L)	24 ± 5	22 ± 6	*<30*
**Pyruvic Transaminase*****(ALT)(GPT)*** (U/L)	19 ± 3	18 ± 4	*<41*
**C Reactive Protein (CRP)** (mg/L)	0.8 ± 0.3	0.7 ± 0.4	*0.0–5.0*

**Table 5 jcm-11-01991-t005:** Seminal parameters investigated at basal conditions (T0) and after three months of nutraceutical administration (T1) (* *p* < 0.05; ** *p* < 0.01).

Semen Parameters	T0 Basal	T1 Post Treatment	*p* Value
**Concentration (×10^6^/mL)**	7.13 ± 4.24	12.24 ± 7.83	*
**Total sperm count (×10^6^/ejaculate)**	21.54 ± 10.15	27.70 ± 14.87	ns
**Vitality (%)**	45.4 ± 8.2	51.2 ± 9.5	*
**Progressive Motility (%)**	22.8 ± 5.9	27.5 ± 6.4	*
**Total motility (%)**	33.6 ± 5.5	40.2 ± 5.8	**
**Morphology (%)**	2.6 ± 1.4	4.2 ± 1.9	**

ns: not significant.

**Table 6 jcm-11-01991-t006:** Reproductive Outcomes of IVF cycle between Group A and Group B (* *p* < 0.05).

Reproductive Outcomes	Group A	Group B	*p* Value
Male patient’s age (years)	34.4 ± 6.8	35.2 ± 6.3	ns
Female patient’age at pick-up (years)	33.7 ± 2.5	34.1 ± 3.2	ns
Number retrieved oocytes	9.8 ± 3.5	9.2 ± 3.3	ns
Number MII oocytes	7.9 ± 2.7	7.5 ± 2.4	ns
Fertilization rate (%)	87.3 ± 15.7	74.3 ± 22.6	*
Pregnancy rate (%)	19.6 ± 3.7	17.2 ± 2.9	ns

## Data Availability

Not applicable.

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
