# Peer review of "Positive Effect of a New Combination of Antioxidants and Natural Hormone Stimulants for the Treatment of Oligoasthenoteratozoospermia"

_jcm, 2022, doi:10.3390/jcm11071991_

Round 1

Reviewer 1 Report

Although the subject is very interesting, there are some methodological errors and inaccuracy in the conclusions:

  1. The number of patients included are really low, about all the number of patients who carried out IVF cycles. This leads to a low statistical power and the presence of a true biological effect of treatment may be masked.
  2. If data on the effect of antioxidants on reproductive outcomes are presented, it is essential to show the characteristics of the women, especially age and diagnosis of infertility.
  3. The discussion should be revised and explain that there are studies that show that antioxidants in unselected patients have not shown any effect.

In addition, the most relevant clinical criterion that should be analyzed after an antioxidant treatment should be the live birth rate and not the sperm quality. The presence of a statistically significant improvement in sperm concentration, motility and morphology does not have to be of clinical importance regarding the fertile potential of an individual. A large number of studies have been conducted examining the ability of different types of antioxidant supplements to improve male reproductive function, however it remains unclear whether male antioxidant therapy before conception can actually improve a couple's chances of become parents.

Therefore, it is to dare to say: 

The effectiveness of this supplementation is definitively demonstrated by the positive effects on the outcomes of assisted reproduction cycles

The only demonstrated effect was fertilization rate, but, only 14 patients is too low, and without any female diagnosis information. 

Author Response

Although the subject is very interesting, there are some methodological errors and inaccuracy in the conclusions:

Q1 – The number of patients included are really low, about all the number of patients who carried out IVF cycles. This leads to a low statistical power and the presence of a true biological effect of treatment may be masked.

A1 – We thank the Reviewer for Her/His revision work. This paper reports the preliminary results of a limited patient cohort. We have chosen to publish our data as “Communication” according to the guidelines of the JCM, because of the limited number of patients enrolled. Our results are preliminary, but still significant. We conducted a detailed selection of our patient cohort (line 97-103) and a proper statistical analysis, that confirms the significance of our results.

Q2 – If data on the effect of antioxidants on reproductive outcomes are presented, it is essential to show the characteristics of the women, especially age and diagnosis of infertility.

A2 – We better detailed the inclusion and exclusion criteria (Line 142-149) and we add the female mean age in the Table 6.

Q3 – The discussion should be revised and explain that there are studies that show that antioxidants in unselected patients have not shown any effect.

A3 – We better detailed the argument in the Discussion paragraph.

Q4 - In addition, the most relevant clinical criterion that should be analyzed after an antioxidant treatment should be the live birth rate and not the sperm quality.

The presence of a statistically significant improvement in sperm concentration, motility and morphology does not have to be of clinical importance regarding the fertile potential of an individual. A large number of studies have been conducted examining the ability of different types of antioxidant supplements to improve male reproductive function, however it remains unclear whether male antioxidant therapy before conception can actually improve a couple's chances of become parents.

A4 - As a measure of the IVF outcome we decide to use the Fertilization rate, and not the Live birth rate, in order to exclude any bias due to female factors (e.g. endometrial receptivity, hormonal and metabolic factors…). The Fertilization rate is an independent predictor of implantation rate.

We are aware that conflicting data exist in the literature, often confounded by significant bias. Our cohort, although limited, is homogeneous in terms of inclusion/exclusion criteria (line 97-103), and the proper statistical analysis confirms the significance of our results. However, we mentioned the limitation of the study in the discussion section, reminding that further studies are needed to improve the power of the study. For this reason, these preliminary results have been submitted as "communication".

Despite the above limitations, we believe that this paper contributes to existing knowledge in this field.

Q5 - Therefore, it is to dare to say: 

The effectiveness of this supplementation is definitively demonstrated by the positive effects on the outcomes of assisted reproduction cycles

The only demonstrated effect was fertilization rate, but, only 14 patients is too low, and without any female diagnosis information. 

A5 – We modified this sentence according to the reviewer suggestion (line 288-290).

Reviewer 2 Report

I read with great interest the study of De Leo et al. entitled: "Positive effect of a new combination of antioxidants and natural hormone stimulants for the treatment of oligoasthenoteratozoospermia",  which was recently submitted in the Journal of Clinical Medicine. 

Male infertility is a medical condition afflicting a large part of the population globally, further causing psychological distress. The authors remarkably present the results of a study demonstrating that the administration of a nutraceutical cocktail can influence androgen production, being considered an effective treatment for OAT. In addition, the authors conveyed the message of the importance of antioxidant supplementation in ameliorating sperm parameters and increasing fertilizing capacity in general.  

The materials and methods section is sufficiently detailed. The infertile patients were carefully evaluated with two consecutive spermiograms prior to study enrollment. Only those with no history of endocrine, metabolic, or anatomical alterations and negative microbiological tests were eligible for the study. 

36 OAT patients were treated daily for 3 months with (Gomotil®, Gofarma, Italy) containing: Inositol, L-Carnitine, Vitamins C, D, E, Coenzyme Q10, and Selenium. Spermiograms, hormonal panels, and hemato-metabolic panels were subsequently investigated.

Moreover, the study was conducted in an Italian Unit of Medically Assisted Reproduction, in Siena, with vast experience in assisted reproductive techniques, which is of great value. Lastly, the results seem promising and should be validated in future trials.

Overall, the manuscript is well-written, clear, concise in the greatest, and should be considered for publication.

Furthermore, minor amendments have to be done before the final acceptance as follows:

Solid evidence demonstrates that OS exerts deleterious effects on healthy spermatozoa via these oxygen-derived free radicals. Nowadays, there is growing awareness that uncontrolled antioxidant therapy is not devoid of risks. More specifically, such an approach is deemed risky, and there is always the danger of diversion to a reductive stress state. Besidesa physiological amount of ROS is always required for the critical processes of capacitation, hyperactivation, acrosome reaction, and sperm–oocyte binding. 

As such, recently, Symeonidis et al. reported the need for Redox Balance. Τhe authors stated that "Mέτρον άριστον," in Ancient Greek, plays a crucial role in every attempt of antioxidant administration. In other words, excellence in using these compounds lies in moderation to avoid the antioxidant paradox phenomenon. 

The authors are advised to add the above remark in the discussion section, thus emphasizing the need for a redox balance guarantee. 

Please add the following citation:

  • Symeonidis EN, Evgeni E, Palapelas V, Koumasi D, Pyrgidis N, Sokolakis I, Hatzichristodoulou G, Tsiampali C, Mykoniatis I, Zachariou A, Sofikitis N, Kaltsas A, Dimitriadis F. Redox Balance in Male Infertility: Excellence through Moderation-"Μέτρον ἄριστον". Antioxidants (Basel). 2021 Sep 27;10(10):1534. doi: 10.3390/antiox10101534. 

Author Response

I read with great interest the study of De Leo et al. entitled: "Positive effect of a new combination of antioxidants and natural hormone stimulants for the treatment of oligoasthenoteratozoospermia",  which was recently submitted in the Journal of Clinical Medicine. 

Male infertility is a medical condition afflicting a large part of the population globally, further causing psychological distress. The authors remarkably present the results of a study demonstrating that the administration of a nutraceutical cocktail can influence androgen production, being considered an effective treatment for OAT. In addition, the authors conveyed the message of the importance of antioxidant supplementation in ameliorating sperm parameters and increasing fertilizing capacity in general.  

The materials and methods section is sufficiently detailed. The infertile patients were carefully evaluated with two consecutive spermiograms prior to study enrollment. Only those with no history of endocrine, metabolic, or anatomical alterations and negative microbiological tests were eligible for the study. 

36 OAT patients were treated daily for 3 months with (Gomotil®, Gofarma, Italy) containing: Inositol, L-Carnitine, Vitamins C, D, E, Coenzyme Q10, and Selenium. Spermiograms, hormonal panels, and hemato-metabolic panels were subsequently investigated.

Moreover, the study was conducted in an Italian Unit of Medically Assisted Reproduction, in Siena, with vast experience in assisted reproductive techniques, which is of great value. Lastly, the results seem promising and should be validated in future trials.

Overall, the manuscript is well-written, clear, concise in the greatest, and should be considered for publication.

Furthermore, minor amendments have to be done before the final acceptance as follows:

Solid evidence demonstrates that OS exerts deleterious effects on healthy spermatozoa via these oxygen-derived free radicals. Nowadays, there is growing awareness that uncontrolled antioxidant therapy is not devoid of risks. More specifically, such an approach is deemed risky, and there is always the danger of diversion to a reductive stress state. Besidesa physiological amount of ROS is always required for the critical processes of capacitation, hyperactivation, acrosome reaction, and sperm–oocyte binding. 

As such, recently, Symeonidis et al. reported the need for Redox Balance. Τhe authors stated that "Mέτρον άριστον," in Ancient Greek, plays a crucial role in every attempt of antioxidant administration. In other words, excellence in using these compounds lies in moderation to avoid the antioxidant paradox phenomenon. 

Q1 - The authors are advised to add the above remark in the discussion section, thus emphasizing the need for a redox balance guarantee. 

A1 – We thank the Reviewer for Her/His important suggestion. We add some sentences in the Discussion paragraph about the redox balance and cited the suggested reference.

Q2 - Please add the following citation:

Symeonidis EN, Evgeni E, Palapelas V, Koumasi D, Pyrgidis N, Sokolakis I, Hatzichristodoulou G, Tsiampali C, Mykoniatis I, Zachariou A, Sofikitis N, Kaltsas A, Dimitriadis F. Redox Balance in Male Infertility: Excellence through Moderation-"Μέτρον ἄριστον". Antioxidants (Basel). 2021 Sep 27;10(10):1534. doi: 10.3390/antiox10101534. 

A2 – Done, the reference was added at line 260 (ref. num. 36).

Round 2

Reviewer 1 Report

Once the review has been carried out by the authors.

Author Response

The review has been carried out.